# The Yin and Yang of Mesenchymal Cells in the Corneal Stromal Fibrosis Response to Injury: The Cornea as a Model of Fibrosis in Other Organs

**DOI:** 10.3390/biom13010087

**Published:** 2022-12-31

**Authors:** Steven E. Wilson

**Affiliations:** The Cole Eye Institute, I-32, Cleveland Clinic, 9500 Euclid Ave, Cleveland, OH 44195, USA; wilsons4@ccf.org

**Keywords:** fibrosis, myofibroblasts, keratocytes, corneal fibroblasts, fibrocytes, TGF beta, epithelial basement membrane, Descemet’s membrane

## Abstract

Mesenchymal cells (keratocytes, corneal fibroblasts, and myofibroblasts), as well as mesenchymal progenitor bone marrow-derived fibrocytes, are the major cellular contributors to stromal fibrosis after injury to the cornea. Corneal fibroblasts, in addition to being major progenitors to myofibroblasts, also have anti-fibrotic functions in (1) the production of non-basement membrane collagen type IV that binds activated transforming growth factor (TGF) beta-1 and TGF beta-2 to downregulate TGF beta effects on cells in the injured stroma, (2) the production of chemokines that modulate the entry of bone marrow-derived cells into the stroma, (3) the production of hepatocyte growth factor and keratinocyte growth factor to regulate corneal epithelial healing, (4) the cooperation with the epithelium or corneal endothelium in the regeneration of the epithelial basement membrane and Descemet’s membrane, and other functions. Fibrocytes also serve as major progenitors to myofibroblasts in the corneal stroma. Thus, mesenchymal cells and mesenchymal cell progenitors serve Yin and Yang functions to inhibit and promote tissue fibrosis depending on the overall regulatory milieu within the injured stroma.

## 1. Introduction

The cornea of the eye (Figure 1) is a uniquely suitable model for studying epithelial–stromal–corneal endothelial–nerve–bone marrow-derived cellular interactions, including those associated with scarring stromal fibrosis, because of its relatively simple and normally avascular structure devoid of hair follicles, glands, and other structures that can obscure cellular interactions. Rabbit and mouse models with a variety of surgical, infectious, and chemical injuries have been used extensively to study the modulation and functions of mesenchymal keratocytes, corneal fibroblasts, and myofibroblasts, as well as bone marrow-derived mesenchymal precursor fibrocytes, in the wound healing response. 

This review will provide a detailed overview of what is currently known about the roles of the different mesenchymal cells and fibrocyte mesenchymal progenitors in the responses to corneal injury. This discussion will rely primarily on using the rabbit model that has been exploited extensively in numerous studies over the past thirty years. The insights that have been gained from studies of the cornea are likely relevant to the roles of mesenchymal cells in all tissues, organs, and organ systems in animals.

## 2. Mesenchymal Cells or Mesenchymal Progenitors

### 2.1. The Keratocyte

The keratocyte is a specialized fibroblast that populates the corneal stroma (Figure 2) that is a neural crest-derived cell [1,2,3]. Keratocytes in situ and in vitro are commonly characterized via their expression of keratocan, along with keratin sulfate, lumican, and collagens, such as collagen type V and collagen type VI [4]. Keratocytes are relatively quiescent cells that rarely proliferate in unwounded corneas. These cells function to maintain the extracellular matrix materials (ECMs) in the stroma that have a precise structure and organization associated with the normal transparency of the corneal stroma [5]. They are also likely involved in cooperative maintaining of the uninjured epithelial basement membrane (EBM) and Descemet’s membrane, along with the corneal epithelium and corneal endothelium, respectively [6,7]. Keratocytes in vitro maintain their phenotype only in special medium without serum [1,2,3]. Keratocytes are variably vimentin-positive in situ, which likely indicates variably low vimentin production relative to corneal fibroblasts or myofibroblasts (Figure 3) [8].

### 2.2. The Corneal Fibroblast

After any type of corneal injury, whether mechanical, chemical, infectious, inflammatory, or surgical, keratocytes in the area of the injury transition into mesenchymal corneal fibroblasts. Corneal fibroblasts express little, if any, keratocan, either in situ or in vitro [1,2,3]. Corneal fibroblasts (Figure 3) develop from resident keratocytes that are stimulated by transforming growth factor (TGF) beta-1 and/or TGF beta-2, platelet-derived growth factor (PDGF), and possibly other yet to be identified cytokines or growth factors [10,11,12,13]. This conversion also involves integrin signaling [10]. There are no specific markers for corneal fibroblasts in situ. Rather, they are identified as being vimentin-positive cells in the corneal stroma that are keratocan-negative and alpha-smooth muscle actin (α-SMA) negative [8,9,14]. 

Corneal fibroblasts, once generated, serve many important wound healing functions. Most of these functions are not performed by the parent keratocytes. It is useful to highlight some of these critical functions performed by corneal fibroblasts (Figure 4).

(1)First, the TGF beta stimulation that drives corneal fibroblast development, also upregulates collagen type IV production by these cells [9,15,16]. This collagen type IV produced in large quantities by corneal fibroblasts serves two major functions. First, collagen type IV binds TGF beta-1 or TGF beta-2 and prevents these growth factors from binding their cognate TGF beta receptors [17,18]. Therefore, the large amounts of collagen type IV produced by the corneal fibroblasts, much of which is not associated with the corneal basement membranes (BM) [9,15,16], down-regulates the effects of TGF beta-1 and TGF beta-2 that enter the stroma from the tears, epithelium, corneal endothelium and/or aqueous humor, and are activated after injury. A major impact of that non-BM collagen type IV, therefore, is to moderate the pro-fibrotic effects of TGF beta by regulating (decreasing) myofibroblast development from the corneal fibroblasts themselves and bone marrow-derived fibrocytes [19,20], the two best-characterized precursors to myofibroblasts in the corneal stroma [1,4,8,9,12,13,14,19,20]. This non-BM collagen type IV production by corneal fibroblasts may be a major determinant of why many relatively low-level corneal injuries, such as abrasions or lower correction photorefractive keratectomy (PRK) surgeries, generate corneal fibroblasts but not myofibroblasts in the corneal stroma [8,21]. With more significant injuries, such as high correction PRKs, alkali burns, or Descemetorhexis, this modulatory system is overwhelmed and large numbers of myofibroblasts develop in the stroma [8,9,14,15,21]. The second function of the collagen type IV produced by corneal fibroblasts is to contribute to EBM and/or Descemet’s membrane regeneration after injury to these important adhesion and regulatory structures in the cornea [9,14]. This regeneration of the BMs is critical to reestablish normal regulation of TGF beta localization to the stroma, eventually trigger myofibroblast apoptosis, and return of the cornea to its normal transparent structure and function [8,9,14].
Figure 4Mesenchymal cells of the injured corneal stroma. The unwounded corneal stroma is populated mostly with keratocytes, with a few nerve cells and resident dendritic cells. At the site of injury, driven by TGF beta-1 and TGF beta-2, along with PDGF and integrin signaling, keratocytes transition into corneal fibroblasts. Corneal fibroblasts that receive sustained and adequate levels of TGF beta-1 and/or TGF beta-2 develop into mature α-SMA-positive myofibroblasts. Another established source of corneal myofibroblasts is fibrocytes which differentiate from bone marrow-derived progenitors that migrate into the cornea from the limbal blood vessels after injury. Fibrocytes that also receive sustained and adequate levels of activated TGF beta-1 and/or TGF beta-2 develop into mature α-SMA-positive myofibroblasts. It remains controversial whether myofibroblasts, once generated, can return to corneal fibroblasts and/or fibrocytes, or rather are always eliminated via apoptosis. Some of the functions of each corneal stromal mesenchymal or mesenchymal progenitor cell type are listed. SAP is serum amyloid protein. SAP most likely functions to inhibit fibrocyte development from precursors but may also directly inhibit fibrocyte differentiation into myofibroblasts [22].
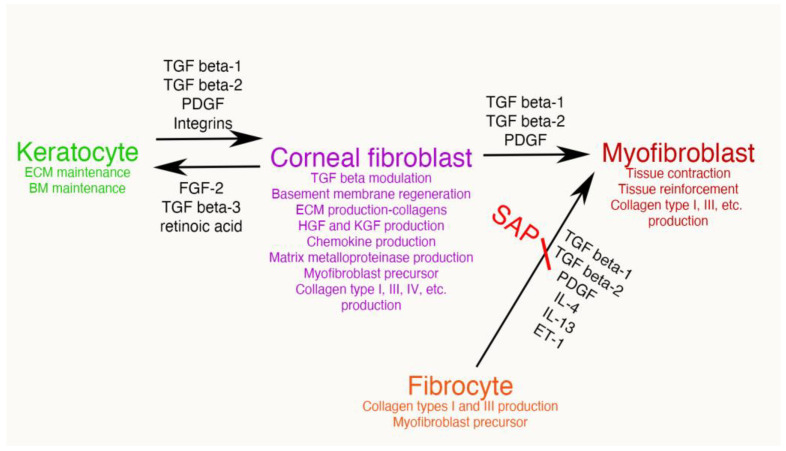



(2)Secondly, corneal fibroblasts produce other BM components that contribute to regeneration of the EBM and/or Descemet’s membrane, such as perlecan, to facilitate regeneration of nascent EBM and Descemet’s membrane after injury [8,14,23]. After the epithelium or endothelium, lays down the earliest components of the EBM or Descemet’s membrane, respectively, including self-polymerizing laminins, then corneal fibroblasts contribute other components [8,9,14]. Defective perlecan insertion into the nascent EBM after injury has been demonstrated in corneas that develop fibrosis compared to those that do not [23].(3)A third major function performed by corneal fibroblasts is the production of growth factors that control the proliferation, migration, differentiation, and apoptosis of the corneal epithelium, including hepatocyte growth factor (HGF) and keratinocyte growth factor (KGF or fibroblast growth factor (FGF)-7) [24,25]. HGF and KGF production by keratocytes is nearly undetectable, but after injury these growth factors are produced at high levels by corneal fibroblasts [25]. Corneal fibroblasts, therefore, are important contributors to the healing and restoration of normal epithelium following injuries.(4)Immediately after corneal injury, fibrocytes, monocytes, polymorphonuclear cells, and other bone marrow-derived cells stream into the corneal stroma from the limbal blood vessels. These cells are attracted by interleukin (IL)-1 alpha, IL-1 beta and other cytokines produced by corneal epithelial and endothelial cells [26]. A fourth important function of corneal fibroblast is the production of chemokines, such as granulocyte-colony stimulating factor (G-CSF), neutrophil-activating peptide (ENA-78), monocyte-derived neutrophil chemotactic factor (MDNCF), and monocyte chemotactic and activating factor (MCAF), after corneal fibroblasts are stimulated by IL-1 or tumor necrosis activating factor alpha (TNFa) [27]. The reference title refers to keratocytes, but the cells studied in vitro were corneal fibroblasts. Therefore, corneal fibroblasts contribute to attracting fibrocytes and immune cells to the injured cornea.(5)A fifth major function of corneal fibroblasts is that they serve as major precursors to stromal myofibroblasts when driven by ongoing and adequate levels of TGF beta-1 and/or TGF beta-2 [1,12,26,28,29,30]. What determines whether a particular corneal fibroblast persists and continues to perform its numerous functions or develops into a myofibroblast in a cornea? That is an area of considerable research interest. Likely, it is the overall milieu that corneal fibroblast finds itself in with regard to the concentrations of activated TGF beta, fibroblast growth factor-2, PDGF, IL-1, and other growth factors and cytokines that determines the fate of that cell at a particular moment during the corneal wound healing response. This is the Yin and Yang of corneal fibroblasts. Although they have an essential role in downregulating myofibroblast development and fibrosis, and non-fibrotic healing of the cornea, they are a primary precursor for myofibroblasts in the cornea.

Many other functions performed by corneal fibroblasts, such as production of metalloproteinases, could be described, but these highlighted roles point to the enormous importance of these cells in the cellular responses to both minor and severe corneal injuries. Even if myofibroblasts are not generated by a particular corneal injury, there is commonly a low level of transient opacity of the corneal stroma due to the greater opacity of the corneal fibroblasts themselves (Figure 1B) [31] and the disordered extracellular matrix they produce prior to resolution of the wound healing response. Once the wound healing response is complete, corneal fibroblasts disappear from the stroma via transition back to keratocytes an/or apoptosis.

### 2.3. The Fibrocyte, a Bone Marrow-Derived Mesenchymal Progenitor

Fibrocytes are bone marrow-derived mesenchymal progenitor cells first described by Bucala and coworkers [32] that co-express hematopoietic stem cell, monocyte lineage, and fibroblast markers [21,23,24,25,26,27,28,29,30,31,32,33,34]. Our lab first demonstrated that bone marrow-derived cells contribute to myofibroblast development in the cornea after injury using chimeric mice that had bone marrow transplants from donor green-fluorescent protein (GFP)-expressing mice after total body irradiation [35]. A subsequent study demonstrated that large numbers of fibrocytes (identified by co-expression of cluster of differentiation (CD)34, CD45, and vimentin) invaded the mouse cornea after irregular phototherapeutic keratectomy (PTK) injury [19]. In this injury model, the majority of fibrocytes died by apoptosis during the early stages of influx into the cornea, but some GFP+ CD45+ α-SMA+ myofibroblasts cells developed from 4 to 21 days after the injury, confirming the development of myofibroblasts from fibrocytes in the cornea.

Recent studies found that fibrocytes are likely not present in the peripheral blood but likely differentiate from a population of CD14+ peripheral blood mononuclear cells [33,36,37,38,39]. These studies suggest that fibrocytes precursors may be present as a small fraction of the CD14+ CD16− subset of human mononuclear cells that bear the CC chemokine receptor (CCR)2 on their surface [32,40,41,42]. When inflammation occurs in an organ, such as the cornea, these CD14+ precursors are released from the bone marrow, enter the peripheral blood, and migrate to the inflamed tissues in a process involving a CCR2-mediated signaling pathway [40,41,42]. Once these CD14+ precursor cells migrate to the inflamed tissue, they are thought likely to differentiate into macrophages or acquire dendritic cellular characteristics [41,42]. An intermediate cellular phenotype may exist [41]. The immature subpopulation of CD14+ mononuclear cells contain many committed precursor cells that can differentiate into a number of differentiated cells, including myoblasts, osteoblasts, chondrocytes, adypocytes, epithelial cells, endothelial cells, neuronal cells and liver cells [43,44]. If this is correct, fibrocytes likely represent an obligate intermediate stage of one of these monocyte lineage precursors that differentiate into mature myofibroblasts, and possibly fibroblasts, at the inflamed tissue site.

Several factors have been shown to modulate the differentiation of fibrocytes and their development into myofibroblasts (and possibly corneal fibroblasts). These include TGF beta-1 (and likely TGF beta-2), as well as PDGF, IL-4, IL-13, and endothelin-1 (ET-1) [33,39,45,46]. Conversely, serum amyloid P (SAP) inhibits the development of fibrocytes from their CD14+ mononuclear precursor cells [38] SAP is a constitutively produced serum protein and it’s been demonstrated to be an active factor in serum that inhibits fibrocyte differentiation from the CD14+ mononuclear cells [37,38].

As will be detailed in the following section, myofibroblasts that develop from bone marrow-derived precursors are not phenotypically identical to myofibroblasts that develop from corneal fibroblasts [47]. Each of these myofibroblasts likely makes unique contributions to the overall fibrosis response to injury.

### 2.4. Myofibroblasts

Mesenchymal myofibroblasts have a critical role in maintaining tissue and organ integrity in response to injury in every organ system of the body. However, they also have a Yin and Yang position in that response to injury because excessive generation of these cells leads to fibrosis, and tissue and organ dysfunction [8,10,12,14,30,47]. Numerous cells, including fibroblasts, fibrocytes, Schwann cells, pericytes, and hepatic stellate cells, can serve as precursors to myofibroblasts [48,49,50,51,52,53,54]. In some organs, epithelial to mesenchymal transition (EMT) or endothelial to mesenchymal transition (EnMT) have been implicated as major contributors to myofibroblast generation in fibrosis [55,56].

Myofibroblasts are classically identified both in vitro and in situ via their expression of α-SMA [47,48,49,50,51,52,53,54]. The numerous precursors of these cells undergo a developmental transition taking days to months, depending on the severity of the tissue injury [8,9,14,15,21,26]. This has been characterized in the cornea in situ where keratocytes, and likely fibrocytes, driven by TGF beta, transition from vimentin+ α-SMA- desmin- precursors to vimentin+ α-SMA+ desmin- intermediate cells to vimentin+ α-SMA+ desmin+ mature myofibroblasts [57]. They function to contract wounds, such as corneal lacerations, and to reinforce damaged tissues, such as after microbial infections, chemical burns or surgical procedures, including Descemetorhexis or high-correction photorefractive keratectomy, in the cornea [8,9,14,15,21,48,49].

Myofibroblasts derived from different precursor cells are likely not identical but contribute in subtly different ways to the overall fibrosis response in a particular organ. This has been demonstrated by detailed proteomic comparisons of myofibroblasts derived from corneal fibroblasts or bone marrow-derived cells by TGF beta-1 induction in cells isolated in parallel from the same individual rabbits [47]. In that study, of the 2329 proteins quantitated, a total of 673 differentially expressed proteins were identified in the two myofibroblast populations. Bioinformatic analysis of differentially expressed proteins with Ingenuity Pathway Analysis implicate progenitor-dependent functional differences in myofibroblasts from different progenitors that likely impact fibrotic tissue development. The results suggested that bone marrow-derived myofibroblasts were more prone to the formation of excessive cellular and extracellular material that is characteristic of fibrosis. 

In a study using corneal fibroblast- and bone marrow-derived cells in mice with green fluorescent protein technology, Singh and coworkers [28] conclusively demonstrated that both precursors developed into α-SMA+ myofibroblasts. This study also showed that the percentage of α-SMA+ myofibroblasts generated from either precursor cell type was higher when both cells were co-cultured together (juxtacrine interactions) as compared to when bone marrow-derived cells and corneal fibroblasts were co-culture in different compartments of a Transwell System (paracrine interactions). Thus, more α-SMA+ GFP-positive myofibroblasts were generated from GFP-positive corneal stromal fibroblasts when GFP-negative bone marrow-derived cells were present and more α-SMA+ GFP-positive myofibroblasts were generated from GFP-positive bone marrow-derived cells when GFP-negative corneal fibroblasts were present.

## 3. Immune Cells

Numerous bone marrow-derived immune cell precursors and immune cells infiltrate the cornea beginning a few hours after injury. These include monocytes, macrophages, neutrophils, T cells and B cells. These cells deal with injuries that involved infectious organism and removal of debris produced by the response to injury. More details in this regard are beyond the scope of this manuscript.

## 4. Mesenchymal Cellular Interactions in the Responses to Corneal Injury

Many different models associated with corneal stromal fibrosis have been explored in rabbits, mice, and other species [8,9,14,15,21,58,59]. Each of these injury models is associated with injury to the epithelium and EBM and/or corneal endothelium and Descemet’s basement membrane. The first observable cellular response with either anterior or posterior corneal injuries is apoptosis of the keratocytes in proximity to the injury [60,61], that is thought to be mediated via activation of the Fas-Fas ligand system [62,63]. At this point, the corneal stroma is populated almost exclusively by keratocytes, and there are no detectible corneal fibroblasts or myofibroblasts [8,9,14,15,21,61,64]. Within hours of the injury, however, residual keratocan-positive, vimentin-positive, α-SMA-negative keratocytes transition to keratocan-negative, vimentin-positive, α-SMA-negative corneal fibroblasts (Figure 5) that begin to proliferate and repopulate the injured stroma [8,9,65,66]. Fibrocytes also invade the corneal stroma beginning a few hours following the injury [19,64].

Thereafter, from days to months after the initial injury [8,9,14,15,21,58,59], depending on the type and severity of the injury (likely related to the levels of activated TGF beta-1 and TGF beta-2 that enter the stroma) keratocan-negative, vimentin-positive, α-SMA-positive myofibroblasts begin to appear in the stroma (Figure 5). These myofibroblasts have been shown to develop from corneal fibroblasts and fibrocytes [19,28] but could potentially also develop from other stromal cells such as Schwann cells or limbal pericytes [49,50,59]. Myofibroblasts commonly increase in numbers and are present in highest density near the anterior and/or posterior corneal surfaces, depending on the original injury, where activated TGF beta-1 and/or TGF beta-2 are likely to be at highest concentrations due to their origin from the tears, injured epithelium, injured corneal endothelium and/or aqueous humor [8,9,14,15,21].

In many corneas, especially after relatively mild injuries, such as corneal abrasions or photorefractive keratectomy for low myopia correction, corneal fibroblasts are generated from keratocytes and fibrocytes invade from the limbal blood vessels, but myofibroblasts are never generated [8,14,58]. In these corneas, the epithelial basement membrane is regenerated within ten days of the injury and the myofibroblast progenitors either undergo apoptosis, or revert to keratocytes, without myofibroblast generation [8,14,19,58]. 

Myofibroblasts, and the scarring fibrosis associated with these cells, persist for months to many years after the original corneal injury, depending on whether or not the basement membranes (EBM and/or Descemet’s membrane) that were injured fully regenerate. If the injured basement membrane(s) regenerate, there is restoration of natural modulation of TGF beta-1 and/or TGF beta-2 entry into the stroma from the tears, epithelium, corneal endothelium and/or aqueous humor, that myofibroblasts depend on for survival [8,9,14,15,21,23,59,64,67]. Thus, the corneal basement membranes (EBM and Descemet’s membrane) via their components perlecan and collagen type IV, are the endogenous modulators of TGF beta entry into the stroma that drives the development of myofibroblasts and their persistence [8,9,14,15,23,58,59]. If the basement membrane(s) fully regenerate at some point, then the requisite supply of TGF beta to the stroma is cut off and the myofibroblasts disappear from the stroma via apoptosis [8,9,14,15,58,59]. This disappearance of myofibroblasts commonly occurs in rabbit corneas several months after high-correction PRK surgery and in human corneas at one to two years after high-correction PRK surgery without the use of intraoperative mitomycin C [8,14,68].

The viability of myofibroblasts in the corneal stroma can also be modulated pharmacologically. Topical, but not oral, losartan has been shown to trigger myofibroblast disappearance, likely via apoptosis, because of losartan’s known TGF beta-signaling inhibition, in both animal models and human case reports of surgical complications [15,21,67,69].

The EBM in the cornea, if it does regenerate through the coordinated actions of epithelium and corneal fibroblasts/keratocytes, typically does so in a spotty distribution across the injured cornea [8,14]. In areas where the EBM regenerated, underlying myofibroblasts are deprived of TGF beta-1 and TGF beta-2 signaling and undergo apoptosis, allowing corneal fibroblasts and keratocytes to repopulate the anterior stroma. These areas are visible as clear “lacunae” in the stromal scarring fibrosis (Figure 1D). Over time, these lacunae tend to enlarge and coalesce until the entire cornea can return to full transparency. This process commonly occurred in patients who had PRK treatments for high myopia prior to the development of intraoperative mitomycin C in PRK surgery [68].

Fibrosis that develops in the cornea is an active tissue populated by myofibroblasts that continue to proliferate (and undergo apoptosis) over time and produce ECM components to maintain the fibrotic tissue for months or even years after the original injury [64]. This likely underlies the efficacy of topical losartan in reducing corneal stromal fibrosis months or years after the initial injury that triggered myofibroblast and fibrosis development [70].

## 5. Conclusions

Mesenchymal cells (keratocytes, corneal fibroblasts, and myofibroblasts) and mesenchymal progenitor fibrocytes are the major contributors to fibrosis in the stroma after injuries to the anterior and/or posterior cornea. TGF beta-1 and TGF beta-2 derived from the tears, epithelium, corneal endothelium, and aqueous humor, depending on the specific injury, along with limited amounts of TGF beta that appears to be produced by bone marrow-derived cells that infiltrate the stroma after injury, are the dominant pro-fibrotic modulators. TGF beta-1/beta-2 modulate the development of corneal fibroblasts from keratocytes, and the development of myofibroblasts from corneal fibroblasts and fibrocytes. The EBM and Descemet’s membrane, via their perlecan and collagen type IV components, serve as major regulators of activated TGF beta localization to the stroma after injury, and the regeneration of these BMs often leads to myofibroblast apoptosis, reorganization of fibrotic tissue, and restoration of normal corneal transparency.

## 6. Patents/Conflicts of Interest

Steven E. Wilson and the Cleveland Clinic submitted a patent on the use of topical losartan and other angiotensin II receptor blockers (ARBs) to prevent and treat corneal scarring fibrosis. 

## Figures and Tables

**Figure 1 biomolecules-13-00087-f001:**
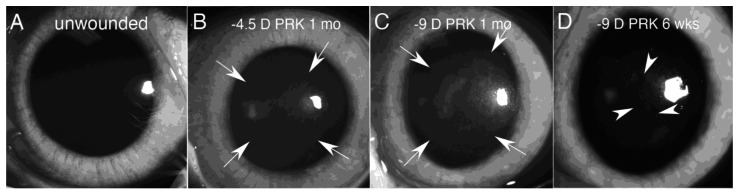
Slit lamp images of rabbit corneas. (**A**) The unwounded cornea is transparent. (**B**) At one month after −4.5 diopter (low myopia correction) PRK (photorefractive keratectomy) there was mild opacity (haze) of the cornea (within arrows). Corneal fibroblasts but no myofibroblasts were present in this cornea with immunohistochemistry (IHC) analysis. (**C**) At one month after −9 diopter (high myopia correction) PRK, there was moderate opacity (haze) of the cornea (within arrows). Myofibroblasts and corneal fibroblasts were now present in the anterior stroma (see Figure 3). (**D**) At six weeks after −9 diopter (high myopia correction) PRK, clear areas referred to as “lacunae” (arrowheads) developed in the moderate opacity where the EBM had fully regenerated and myofibroblasts had undergone apoptosis. Over time, these lacunae will become more numerous, enlarge, and coalesce, until complete transparency is restored to the cornea. Mag 15×.

**Figure 2 biomolecules-13-00087-f002:**
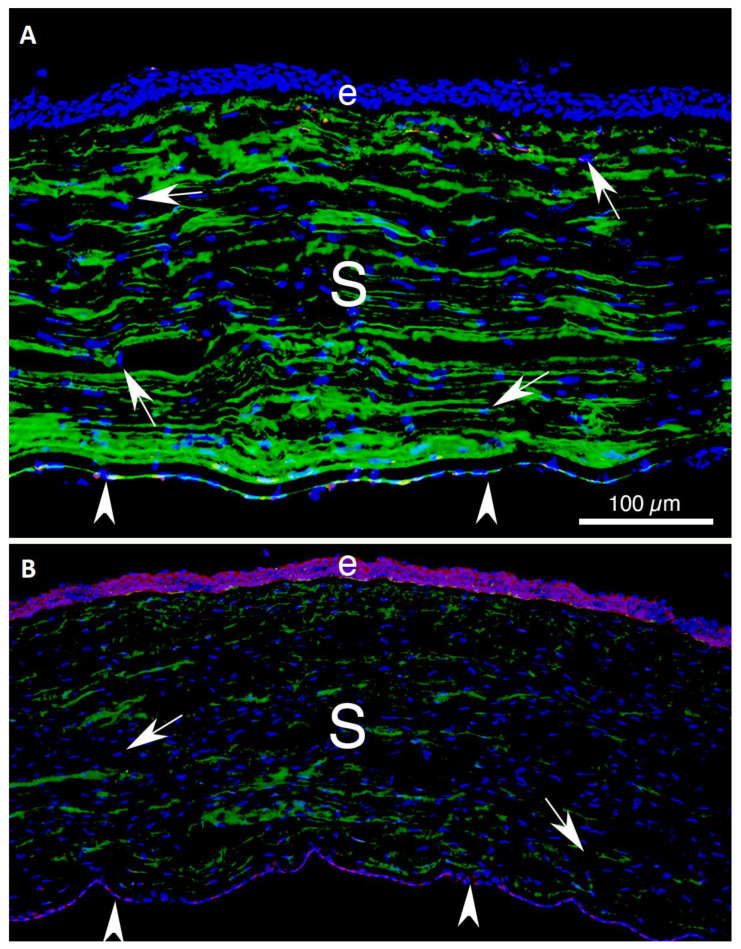
Structure of the cornea. (**A**) A transverse section of an unwounded rabbit cornea that had immunohistochemistry (IHC) for keratocan (green) and alpha-smooth muscle actin (α-SMA) (red), along with DAPI (blue) stained nuclei. e is the epithelium. Arrowheads indicate the corneal endothelium that lines the posterior surface of the cornea. The central stroma (S) comprises approximately 95% of the corneal thickness and is populated primarily with keratocytes (arrows), although a few resident immune cells and nerves are also present (not shown). No *α*-SMA cells were detected. (**B**) An unwounded rabbit cornea that had IHC for TGF beta-1 (pink) and keratocan (green). Keratocan is present throughout the (S) and associated with keratocytes (arrows). Note that in the unwounded cornea, TGF beta-1 is detected in the epithelium (e) and the corneal endothelium (arrowheads), with little, if any, detected in the corneal stroma. Reprinted with permission from Ref. [8]. Copyright 2021 Elsevier.

**Figure 3 biomolecules-13-00087-f003:**
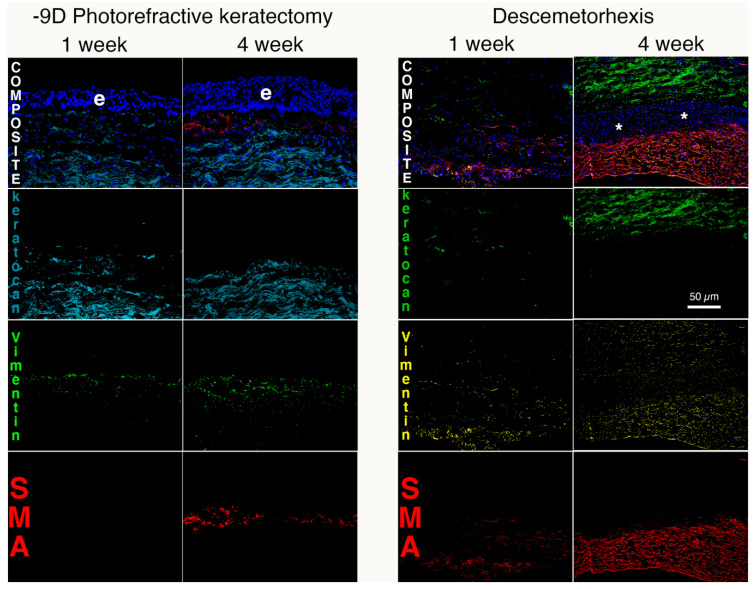
Mesenchymal keratocytes, corneal fibroblasts, fibrocytes and myofibroblasts in the response to corneal injury. Triplex IHC for keratocan, vimentin and α-SMA after photorefractive keratectomy (PRK) for high myopia without mitomycin C treatment or Descemetorhexis (removal of an 8 mm diameter circle of Descemet’s membrane and associated corneal endothelium. At one week after PRK, keratocan-positive keratocytes are diminished in the anterior stroma beneath the epithelium (e) and there are increased vimentin-positive, α-SMA-negative cells that are primarily corneal fibroblasts and fibrocytes (compare to Figure 2A). No α-SMA-positive myofibroblasts are present. At four weeks after PRK, the anterior stroma is populated by a mixture of α-SMA-positive, vimentin-positive myofibroblasts and α-SMA-negative, vimentin-positive, keratocan-negative corneal fibroblasts and, likely, fibrocytes. Note the epithelial (e) hyperplasia. At one week after Descemetorhexis (only the posterior stroma is shown), the injury is so severe that some α-SMA-positive myofibroblasts have already been generated and the more anterior stroma is populated with -SMA-negative, vimentin-positive, keratocan-negative corneal fibroblasts and likely fibrocytes. By four weeks after Descemetorhexis, when the corneal endothelium and Descemet’s membrane have yet to regenerate, the most posterior stroma is occupied by a dense population of α-SMA-positive myofibroblasts. Between the myofibroblast layer, and the more anterior keratocan-positive keratocyte layer, there is a layer of vimentin-positive, α-SMA-negative, keratocan-negative cells (*) that are corneal fibroblasts and residual fibrocytes. Reprinted with permission from Ref. [8]. Copyright 2021 Elsevier. Reprinted with permission from Ref. [9]. Copyright 2021 Elsevier.

**Figure 5 biomolecules-13-00087-f005:**
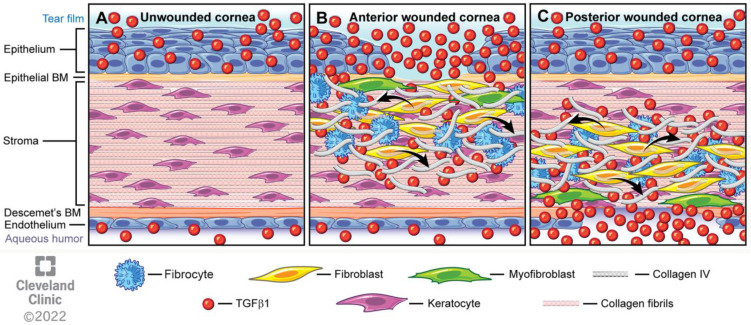
Schematic figure highlighting the roles of keratocytes, corneal fibroblasts, fibrocytes, and myofibroblasts in the cornea, along with the critical roles of the corneal basement membranes (the epithelial basement membrane and Descemet’s membrane). (**A**) In the unwounded cornea, the corneal stroma is populated with keratocytes. The epithelial BM and Descemet’s BM inhibit TGF beta-1 (and TGF beta-2, not shown) passage into the stroma from the tear film, epithelium, corneal endothelium, and aqueous humor; (**B**) After an anterior wound, such as trauma or photorefractive keratectomy, to the epithelium, epithelial BM, and stroma, activated TGF beta-1 (and TGF beta-2), along with other growth factors (platelet-derived growth factor and fibroblast growth factor-1, for example), enter the stroma and bind receptors on anterior keratocytes. Keratocytes then transition to corneal fibroblasts. Corneal fibroblasts, along with infiltrating bone-marrow-derived fibrocytes, begin a one-week to several-month transition to fibrosis-producing myofibroblasts. TGF beta-1 and TGF beta-2 also trigger the production of COL IV by corneal fibroblasts (arrows). COL IV binds TGF beta-1 and TGF beta-2, impedes TGF beta binding to the cognate TGF beta receptors, inhibits the mitosis and further development of corneal fibroblasts, and promotes the apoptosis of myofibroblast precursor cells and myofibroblasts themselves that are dependent on adequate ongoing levels of TGF beta-1 and/or TGF beta-2 for survival. The development of fibrosis is halted by the regeneration of the normal, mature epithelial BM that once again restricts passage of TGF beta-1 and TGF beta-2 into the stroma, leading to apoptosis of the myofibroblast precursor cells and myofibroblasts. After more severe injuries, established myofibroblasts and fibrosis can resolve after late regeneration of the epithelial BM, but may persist in some eyes; (**C**) Posterior wounds, such as surgical Descemetorhexis (removal of Descemet’s membrane and associated corneal endothelial cells), that injure the endothelium and Descemet’s BM, facilitate entry of TGF beta-1 and TGF beta-2 (not shown) into the stroma from the aqueous humor and residual corneal endothelial cells. This triggers similar corneal fibroblast transition, production of COL IV by corneal fibroblasts (arrows), and the development of myofibroblasts in the posterior cornea that is halted by the regeneration of the corneal endothelium and Descemet’s BM through the coordinated efforts of corneal endothelial cells and corneal fibroblasts. The actual number and variety of immune cells in the injured corneas is much greater than depicted. Reprinted with permission from Ref. [16]. Copyright 2022 Elsevier.

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
