# Peer review of "The Yin and Yang of Mesenchymal Cells in the Corneal Stromal Fibrosis Response to Injury: The Cornea as a Model of Fibrosis in Other Organs"

_biomolecules, 2022, doi:10.3390/biom13010087_

Round 1

Reviewer 1 Report

In this review article, Dr. Wilson provides an overview of the roles of different mesenchymal cells and its progenitors in the process of corneal healing and fibrosis.

The manuscript is very well written, but the section headings need to be revised. Some of the numbers are duplicated (2. The keratocyte line 42; 2. The corneal fibroblast line 77; 4. Myofibroblasts line 212; 4. Mesenchymal cellular interactions in the responses to corneal injury line 245). Please consider using second-level headings for each one of the mesenchymal cell types and fibrocyte mesenchymal progenitors, e.g., 2.1 Keratocytes, 2.2 Fibroblasts, etc ...

In Figure 4, since you have specified the types of collagens (I and II) produced by fibrocytes, I think you should also specify the types of collagens produced by fibroblasts, including collagen type IV, due to its importance in the corneal healing process.

Even if it is out of the scope of this manuscript, I think you should consider adding a few sentences discussing the role of other cells, such as neutrophils, in the response to corneal injury.

Author Response

Reviewer 1

  1. The manuscript is very well written, but the section headings need to be revised. Some of the numbers are duplicated (The keratocyte line 42; 2. The corneal fibroblast line 77; 4. Myofibroblasts line 212; 4. Mesenchymal cellular interactions in the responses to corneal injury line 245). Please consider using second-level headings for each one of the mesenchymal cell types and fibrocyte mesenchymal progenitors, e.g., 2.1 Keratocytes, 2.2 Fibroblasts, etc ...

Thank you. I have corrected this error in the revised manuscript.

  1. In Figure 4, since you have specified the types of collagens (I and II) produced by fibrocytes, I think you should also specify the types of collagens produced by fibroblasts, including collagen type IV, due to its importance in the corneal healing process.

This has been added for both corneal fibroblasts and myofibroblasts

  1. Even if it is out of the scope of this manuscript, I think you should consider adding a few sentences discussing the role of other cells, such as neutrophils, in the response to corneal injury.

I added another section “3. Immune cells” and briefly noted their functions in corneal wound healing

Reviewer 2 Report

This well-written review effectively summarizes the role of mesenchymal cells in the stromal fibrotic response of the cornea. The information is presented in a logical order and the figures help illustrate the key points discussed by the review.

Minor Comments:

1.    Abstract: lines 11-18 list all of the anti-fibrotic functions of the corneal fibroblasts. It may be helpful to number each function or break up this sentence into more than one sentence.

2.    Figure 2: Please include labeling on the figure panels (a & b).

3.    Figure 4: Models the mesenchymal cells of the injured corneal stroma and their key functions.

a.    It may be useful to number the functions to parallel the main text

b.    In the figure, it is difficult to understand the role of SAP for fibrocyte to myofibroblast transition. It would be useful to illustrate as a block. How does SAP block fibrocyte to myofibroblast transition? Does it involve impacting TGFB, PDGF, IL-4, IL-13 or ET-1?

c.     Please define abbreviations in the legend (i.e. SAP)

4.    Is anything known about the potential juxtactrine interactions between corneal fibroblasts and bone marrow-derived cells, which enables these cells to increase the percentage of αSMA+ myofibroblasts generated?

Author Response

  1. Abstract: lines 11-18 list all of the anti-fibrotic functions of the corneal fibroblasts. It may be helpful to number each function or break up this sentence into more than one sentence. 

I’ve numbered the functions in the sentence as suggested.

  1. Figure 2: Please include labeling on the figure panels (a & b).

a and b were already in the upper left hand corner of each panel.

  1. Figure 4: Models the mesenchymal cells of the injured corneal stroma and their key functions. 
  2. It may be useful to number the functions to parallel the main text

            I tried to do this and it just looked sloppy.

5. In the figure, it is difficult to understand the role of SAP for fibrocyte to myofibroblast transition. It would be useful to illustrate as a block. How does SAP block fibrocyte to myofibroblast transition? Does it involve impacting TGFB, PDGF, IL-4, IL-13 or ET-1? 

I added what is known about SAP inhibition of fibrocyte development from precursors and possibly direct involvement in fibrocyte transition to myofibroblasts to the figure legend, but the specific modulators involved in these inhibitions is unknown. I added reference 71.

6. Please define abbreviations in the legend (i.e. SAP)

I defined the SAP abbreviation in the figure legend.

7. Is anything known about the potential juxtactrine interactions between corneal fibroblasts and bonemarrow-derived cells, which enables these cells to increase the percentage of αSMA+ myofibroblasts generated?

No, we have not looked into this further as of yet. But we plan to in experiments later in 2023.